# Behavioral Assessment Reveals GnRH Immunocastration as a Better Alternative to Surgical Castration

**DOI:** 10.3390/ani14192796

**Published:** 2024-09-27

**Authors:** Liuxia Lin, Mengsi Xu, Jian Ma, Chunmei Du, Yaxin Zang, Amei Huang, Chen Wei, Qinghua Gao, Shangquan Gan

**Affiliations:** 1College of Life Science, Tarim University, Alaer 843300, China; 107572021315@stumail.taru.edu.cn (L.L.); 15938368286@163.com (A.H.); 2College of Coastal Agricultural Sciences, Guangdong Ocean University, Zhanjiang 524088, China; crazyma0411@163.com (J.M.); duchunmeim@163.com (C.D.); weichenwjf@126.com (C.W.); 3State Key Laboratory of Sheep Genetic Improvement and Healthy Production, Xinjiang Academy of Agricultural and Reclamation Sciences, Shihezi 832000, China; xumengsi100@163.com; 4College of Animal Science, Xinjiang Agricultural University, Urumqi 830052, China; zangyaxin163@163.com; 5College of Animal Science and Technology, Tarim University, Alaer 843300, China

**Keywords:** GnRH immunocastration, reproductive suppression, animal welfare, emotion, social interaction, learning and memory

## Abstract

**Simple Summary:**

Surgical castration is a common practice for controlling animal reproduction. However, surgical castration can cause pain and stress and may even lead to psychological harm. As an alternative, GnRH vaccines can lower androgen levels without surgery. In this study, we compared the effects of a newly developed GnRH vaccine and surgical castration in rats. Both methods effectively suppressed sexual behavior, but surgical castration resulted in significant depression and social withdrawal, while the vaccinated rats did not exhibit these behavioral issues. This suggests that GnRH immunocastration may be a more humane method that better aligns with animal welfare standards.

**Abstract:**

Castration is often employed in animal management for reproductive control. However, it is important to evaluate its impact on animal welfare. In this study, we developed rat models for both surgical (*n* = 6) and GnRH immunocastration (*n* = 6) to assess the effects of these castration methods on physiological and behavioral characteristics. The novel GnRH-based vaccine significantly increased serum GnRH antibody levels and drastically reduced testosterone, with the testes shrinking to one-fifth the size of those in the control group, thereby halting spermatogenesis at the secondary spermatocyte stage. Behavioral evaluations demonstrated that sexual behavior was significantly suppressed in both surgically and immunologically castrated groups compared to the control, confirming the effectiveness of both methods. However, psychological tests revealed significant signs of depression and social deficits in the surgically castrated group, whereas the behavior of the GnRH-immunocastrated group did not significantly differ from the control. Furthermore, no significant differences in learning and memory were observed among the three groups in the water maze test. Compared to surgical castration, GnRH immunocastration offers effective results and better animal welfare, providing a more humane alternative for livestock management.

## 1. Introduction

Animal reproductive control is essential in various domains, including pet management, livestock farming, and wildlife conservation. In companion animals, castration offers numerous benefits, such as population control and the mitigation of undesirable behaviors like aggression and territorial marking. Neutered animals generally exhibit increased docility, are easier to manage, and have a reduced risk of certain health conditions, such as prostate disease and pyometra [1,2]. In livestock farming, male animals are primarily raised for meat production. Castration is a common practice to effectively reduce the production of androstenone and skatole, compounds responsible for “male taint,” thereby improving meat flavor. Additionally, castration promotes fat deposition, enhances meat tenderness, and slightly lowers meat pH levels, contributing to increased market value [3]. In wildlife conservation, castration serves as a strategic tool for population control, particularly in polygynous species where one male can impregnate multiple females. This method helps manage species numbers and mitigates the ecological strain caused by overpopulation [4,5].

Although traditional surgical castration is often used to control the reproductive capabilities of male animals, it is an invasive method that inevitably impacts animal welfare. Surgical castration may cause animals to endure pain and experience prolonged stress responses, which in turn can affect their behavior and emotions, such as increased anxiety and depression [1].

Immunocastration, as an emerging non-invasive reproductive control technology, has garnered attention for its potential positive impact on animal welfare. Currently available immunocastration vaccines target substances like gonadotropin-releasing hormone (GnRH) [2], follicle-stimulating hormone (FSH) [3], gonadotropin receptors [4,5], GnRH receptor (GnRHR) [6], sperm-specific proteins [7], and zona pellucida glycoproteins [8]. Among these, gonadotropin-releasing hormone (GnRH), situated at the apex of the hypothalamic-pituitary-gonadal (HPG) axis, serves as the ultimate signaling molecule in regulating reproduction in mammals and is considered the optimal target hormone for reproductive immunoregulation [9]. Active immunization with a GnRH vaccine can induce an autologous immune response against GnRH, thereby interfering with the HPG axis and hindering the synthesis and secretion of pituitary gonadotropins and gonadal steroids, effectively suppressing the production of sex hormones, gonadal development, and the expression of sexual behaviors [10]. Currently, despite extensive studies on the impacts of GnRH vaccines on reproductive hormone levels, as well as the structure and functionality of reproductive organs [11,12], the understanding of the broad behavioral and psychological effects of this immunological intervention is still relatively limited. The neuroendocrine system intricately connects reproductive hormones with brain regions that are involved in a wide array of behaviors [13]. Specifically, testosterone, the primary male hormone, influences not only sexual behavior in the brain but also participates in complex neurobiological processes including learning and memory, emotional regulation, and social interactions [14,15]. Therefore, it is plausible to hypothesize that GnRH vaccines, by modifying the GnRH pathway, may inadvertently impact these non-reproductive behaviors.

Existing commercial GnRH vaccines typically utilize chemically synthesized oligopeptides conjugated with carrier proteins, which present several limitations, including high production costs, significant side effects, limited immunogenicity, short immune duration, and the necessity for frequent booster doses, thereby hindering their widespread application [16,17]. To address these challenges, we developed a novel recombinant GnRH vaccine. This vaccine has been optimized through techniques such as site-directed mutagenesis, peptide insertion, and sequence repetition, resulting in a significant enhancement of its immunogenicity. Furthermore, it is produced using an Escherichia coli expression system, which improves production efficiency and reduces costs, facilitating broader application. This vaccine operates on the same mechanism as commercial GnRH vaccines, with enhanced immunogenicity being the sole intended modification. In this study, we administered this novel vaccine to sexually mature SD male rats to conduct an exhaustive evaluation of the impact on reproductive physiology, sexual behaviors, and additional behavioral parameters. A meticulously designed experiment suite was employed, encompassing the Morris water maze to assess cognitive functions related to learning and memory [18], the sucrose preference test and the open field test for evaluating affective states [19], and the three-chamber social interaction test (3 CST) to monitor social behaviors [20]. These experiments are intended to provide a preliminary insight into whether GnRH immunocastration extends beyond reproductive capacities to affect other critical behavioral and psychological welfare metrics. Such a comprehensive evaluation is essential to validate the GnRH vaccine as a viable non-surgical alternative for castration, ensuring the safety and the implications for animal welfare in applied settings.

## 2. Materials and Methods

### 2.1. Preparation of GnRH Engineered Vaccine

To enhance the immunogenicity of the GnRH vaccine, a strategic optimization of its amino acid sequence was undertaken. Employing a multifaceted approach, we incorporated techniques such as site-directed mutagenesis, peptide insertion, and sequence duplication, focusing on critical epitopes within the mature GnRH peptide to maximize immunogenic responses. Given the proprietary nature of this work, specific details on the sequence alterations remain confidential pending patent approvals. The optimized peptide sequence was reverse-translated to a synthetic gene, codon-optimized for Escherichia coli expression. This gene was cloned into the advanced prokaryotic expression vector, pET-32a(+), which facilitates enhanced solubility and improved yield of heterologous proteins. Post-cloning, the construct was confirmed via high-throughput sequencing and restriction enzyme analysis. Transformation was carried out in E. coli BL21(DE3) competent cells, a strain selected for its robust protein production capabilities. Protein expression was induced under controlled conditions using IPTG, optimizing both the temperature and inducer concentration to refine the expression profile and minimize inclusion body formation. The lysate was subjected to ultrasonication under optimized conditions to ensure efficient cell disruption while preserving protein integrity. Purification of the recombinant GnRH was conducted using a high-affinity nickel-nitrilotriacetic acid (Ni-NTA) chromatography system, designed to achieve high purity levels. The protein’s integrity and purity were assessed through SDS-PAGE, complemented by Western blot analysis using a GnRH-specific antibody. Finally, the purified protein was dialyzed against a physiologically relevant buffer to promote correct folding. The protein concentration was quantified using the bicinchoninic acid (BCA) protein assay, ensuring accurate dosage formulation. The final vaccine preparation involved mixing the purified, refolded protein with a scientifically selected adjuvant, enhancing the immunogenic potential when administered in subsequent animal trials. This comprehensive approach ensures that the GnRH vaccine is not only potent in terms of eliciting a targeted immune response but also adheres to the highest standards of biotechnological innovations in vaccine development.

### 2.2. Experimental Design and Grouping

To evaluate the behavioral impacts of different castration methods and the physiological saline control group on rats, the construction of animal models and behavioral identification indicators for the intact physiological saline control group (IC), GnRH Vaccine Immunocastration Group (IM), and Surgical Castration Group (SC) are shown in Figure 1. Our study employed the healthy male Sprague–Dawley rats aged 7–8 weeks and weighing between 180 and 220 g. All animals were confirmed to meet specific pathogen-free conditions. The rats were randomly assigned to one of three experimental groups, with each group comprising six rats (*n* = 6). The groups were designated as follows.

#### 2.2.1. Surgical Castration Group (SC)

At the start of the study (week 0), rats in this group underwent bilateral orchiectomy under anesthesia induced by intraperitoneal injection of Fluamine (0.6 mL/kg) and Diazepam (2.5 mg/kg). The animals received post-operative analgesia with carprofen (5–10 mg/kg, administered subcutaneously every 12–24 h) to effectively manage pain and inflammation following the surgery. Temperature support was provided using heating pads or thermal blankets to maintain normal body temperature during the early post-operative period, preventing hypothermia or overheating. Wound sites were monitored at least twice daily for signs of infection, such as redness, swelling, or discharge. Any potential complications were promptly addressed. The animals had free access to food and clean water supplemented with an appropriate amount of vitamin C. Their overall condition, including activity levels and food intake, was closely monitored to ensure a smooth recovery.

#### 2.2.2. GnRH Vaccine Immunocastration Group (IM)

Rats in this group received an intramuscular injection of a genetically engineered vaccine containing 200 µg recombinant GnRH protein at week 0 and week 4. Injections were alternated between the left and right hind limb quadriceps to minimize muscle irritation at the injection sites.

#### 2.2.3. Intact Control Group (IC)

Rats in this group received placebo injections of the same volume and composition as the IM group at the corresponding time points; however, the placebo did not contain any active GnRH components.

### 2.3. Ethical Note

Animal welfare was monitored daily throughout the study, and all procedures were approved by the Experimental Animal Ethics Committee of Tarim University (ethics number No. DTU 20230126). The rats involved in this research were housed in groups of three per cage, with all rats in each cage belonging to the same treatment group, in cages sized 50 × 38 × 15 cm placed in a clean and dry temperature-controlled animal facility (22 ± 2 °C) with a 12 h light/12 h dark cycle. Throughout the experiment, all rats had ad libitum access to food and water.

### 2.4. Measurement of Anti-GnRH Antibody and Testosterone

Blood samples were collected from the rats at weeks 0, 2, 4, 6, 8, 10, and 12 for antibody titer and testosterone concentration analyses. The method for determining the serum anti-GnRH antibody titers was adapted from Fromme et al. [21] using an indirect enzyme-linked immunosorbent assay (ELISA). Initially, GnRH recombinant protein (1 μg/mL) was coated onto a 96-well microplate. Non-specific binding sites were blocked using 5% non-fat milk powder. Serum samples were then diluted 1:500 and added to the coated wells. Detection was carried out using horseradish peroxidase-conjugated anti-rat IgG (HRP) at a dilution ratio of 1:50,000. The reaction was developed by adding TMB (tetramethylbenzidine) substrate and subsequently terminated with 1 M sulfuric acid. The optical density (OD) of each well was measured at a wavelength of 450 nm to quantify the antibody titers.

Serum testosterone concentrations were quantified by commercial ELISA kits for rat, according to the manufacturer’s instructions (Jonln Co., Ltd., Shanghai, China). The sensitivity of the assay was 0.06 ng/mL, with intra- and interassay CVs of 3.3% and 4.6%, respectively.

### 2.5. Testicular Weighing and Histomorphological Observation

At the conclusion of the experiment in week 14, all rats were euthanized using sodium pentobarbital (50 mg/kg, intraperitoneal injection). The testes were excised rapidly, rinsed with saline to remove blood and attachments, and then weighed to determine their wet weight. The testicular coefficient (TC) was calculated as follows: (TC = testicular weight (g)/body weight (g) × 100%). Following the method described by Slaoui [22], testicular tissues were prepared for histological examination. The specific steps included fixation in 10% neutral buffered formalin, routine paraffin embedding, sectioning, hematoxylin and eosin (H&E) staining, and observation under a light microscope. The sections were cut to a thickness of 5 μm. After staining, the sections were dehydrated and mounted for morphological observation and assessment under a light microscope.

### 2.6. Behavioral Assessment

To assess the fertility and sexual behavior of immunized SD male rats, mating trials were conducted at week 12 post-immunization with both the IC and IM groups. Given that surgical castration is known to suppress sexual behavior in rats, we focused our observations on the immunized and control groups, omitting the surgically castrated group.

Following the mating trials, in the 13th week after the start of the experiment, all three groups (IC, IM, and SC) underwent a series of behavioral tests. These included the Morris Water Maze (MWM) to assess spatial learning and memory, the sucrose preference test (SPT) to evaluate anhedonia, the Open Field Test (OFT) to measure locomotor activity and anxiety-like behavior, and the Three-Chamber Sociability Test (3CST) to examine social interaction.

#### 2.6.1. Observation of Sexual Behavior

To assess the fertility and sexual performance of SD male rats following immunization, mating trials with the IC and IM groups were conducted in the 12th week after the start of the experiment. Numerous studies have confirmed that surgical castration inhibits sexual behavior in rats [23], so we did not observe sexual behavior in the surgically castrated group based on this consideration. Each evening at 23:00, male rats were paired with estrous females (determined by the presence of a large number of flattened keratinized cells in vaginal smears) in a ratio 1:1. The entire mating process was recorded using infrared surveillance to ensure continuous monitoring. Behavioral indicators of sexual activity were evaluated by reviewing the recorded footage according to the criteria established by Liu et al. [24] (Table 1). All behavioral data were independently collected by two trained scorers, and the average of their scores was taken as the final result. In cases of disagreement, a third scorer was appointed to make the final decision.

#### 2.6.2. Learning and Memory Capacity Test

The Morris Water Maze (MWM) involves a circular pool (120 cm diameter, 60 cm depth) with a white escape platform (10 cm diameter). Before testing, the rats practiced locating a visible escape platform from four quadrants, guided to it if not found within 120 s, to acclimate. The main test consisted of two parts: the Spatial Navigation Test and the Probe Trial. The Spatial Navigation Test was conducted over five days with four trials per day. The platform was hidden 1 cm below the water surface in the second quadrant. The rats were placed in the water at the midpoint of the periphery of each quadrant, facing the pool wall, in a fixed sequence. The time taken for the rats to climb onto the platform after entering the water was recorded as the escape latency, with a 120 s limit per trial. If unsuccessful, the rats were guided to the platform and allowed to stay there for 10 s to learn the spatial cues. In the Probe Trial, the platform was removed. The rats were monitored for their ability to recall the platform’s location, assessing the time spent and number of crossings in the target quadrant over 120 s.

#### 2.6.3. Sucrose Preference Test

The sucrose preference test (SPT) is a widely utilized method for evaluating emotional preferences in rats [25]. This test encompasses both a training and a testing phase. Initially, rats are acclimated to a 1% sucrose solution, provided in two bottles, to establish a preference for sucrose water. One bottle is replaced with tap water the following day; to mitigate location bias, the positions of the sucrose solution and tap water bottles are switched every 8 h. During the testing phase, after 24 h of fasting, rats are given unrestricted access to both water and sucrose solution for 24 h. The consumption of each liquid is recorded, and the preference for sucrose is calculated as the ratio of sucrose intake to the total liquid intake, expressed as a percentage.

#### 2.6.4. Anxiety and Exploratory Behavior Assessment

The Open Field Test (OFT) is a common experimental method used in behavioral neuroscience to assess levels of anxiety, exploratory behavior, and overall activity in rats, The test was conducted in a 100 cm × 100 cm square arena with a height of 40 cm to assess anxiety-like behaviors in rats. The arena was cleaned with 75% ethanol before and after each test to eliminate odor residues. A video tracking system recorded the total distance moved by the rats within 5 min as well as the distance moved and time spent in the center of the arena, as indicators of anxiety-related behavior.

#### 2.6.5. Social Behavior and Social Preferences

The Three-Chamber Sociability Test, detailed by Xiao et al. [26], is used to evaluate social behavior and preferences in rats within a rectangular arena divided into three compartments. The test consists of three continuous stages: environmental adaptation, social interaction, and social novelty preference. In the Environmental Adaptation stage, rats are placed in the middle compartment and allowed to freely explore the arena for 5 min to acclimate. In the Sociability Test stage, a novel “stranger” rat is introduced into one of the side compartments, while the other side remains empty. The test subject is again allowed to explore for 5 min, during which social interest towards the stranger rat is evaluated. In the Social Preference Test stage, a second novel “stranger” rat is introduced into the previously empty compartment, while the first rat remains in the other side, now considered the “familiar” rat. The subject’s preference for social novelty is assessed by allowing free exploration for an additional 5 min. Rats freely explore for assessment of social interest and recognition, monitored via a video tracking system documenting their movements and time spent across phases.

### 2.7. Data Analysis

All analyses were performed in IBM© SPSS© Statistics (version 26.0 for Windows; IBM, Armonk, NY, USA). The normality of all data was first assessed using the Shapiro–Wilk test. Normally distributed data (*p >* 0.05) were presented as mean values with standard deviations (SD), while non-normally distributed data were reported as medians with interquartile ranges (IQR). For comparisons between two groups, such as testicular indices and sexual behavior data, independent samples t-tests were used for normally distributed data, and Mann–Whitney U tests were used for non-normally distributed data. For multiple group comparisons, including datasets like MWM, SPT, and OFT, ANOVA was applied to data that were both normally distributed and had equal variances followed by post-hoc tests with Bonferroni corrections. For data with unequal variances, Welch’s ANOVA was used. Non-parametric data were analyzed using the Kruskal–Wallis H test, with subsequent post-hoc analysis by Dunn’s test. For continuous sampling data, such as body weight, serum GnRH antibody titers, testosterone levels, and 3CST datasets, repeated measures ANOVA was used for normally distributed data, with Greenhouse–Geisser correction applied when the sphericity assumption was violated. Post-hoc tests were adjusted using Bonferroni corrections. Statistical significance was determined at a level of *p* < 0.05 for all analyses.

## 3. Results

### 3.1. Development and Assessment of High-Efficiency Vaccine-Induced Castration Models in Rats

As illustrated in Figure 2A, antibody levels in the immunized (IM) and control (IC) groups exhibited significantly different patterns over time (F(1.243, 4.973) = 31.413, *p =* 0.002) and differed significantly between groups (F(1, 4) = 477.274, *p* < 0.0001). At initial immunization (week 0), both groups showed low antibody levels without significant difference. However, at 2 weeks post-immunization, the IM group’s antibody levels increased rapidly (mean ± SD = 1.371 ± 0.533), significantly exceeding the IC group (mean ± SD = 0.029 ± 0.007, *p =* 0.012). After the booster immunization, the IM group showed peak antibody levels at week 6 (mean ± SD = 2.704 ± 0.065), significantly higher than the week 0 levels (*p* < 0.0001). Subsequently, despite a slight decline, the IM group’s antibody levels remained significantly higher than the IC group at week 8, week 10, and week 12 (*p* < 0.0001 for all time points). These results indicate that the two immunizations successfully induced a persistent antibody response, particularly pronounced after the second immunization (Figure 2A).

The testosterone levels in the serum exhibited an opposite pattern to the GnRH antibody levels. Repeated measures ANOVA revealed significant differences in overall testosterone levels among IC, IM, and SC groups throughout the experiment (F(2, 6) = 646.530, *p* < 0.0001), significant changes in rat serum testosterone levels across different time points (F(6, 36) = 70.128, *p* < 0.0001), and a significant interaction effect in testosterone level trends among the three groups over time (F(12, 36) = 27.107, *p* < 0.0001). At the experiment’s initiation (week 0), there were no statistically significant differences in testosterone levels among the three groups (*p >* 0.05). By week 2, due to orchiectomy, the SC group’s testosterone levels dropped sharply compared to week 0 (SC week 2: mean ± SD = 0.95 ± 0.11 ng/mL vs. SC week 0: mean ± SD = 4.77 ± 0.51, *p =* 0.009). After initial GnRH vaccine immunization, by week 2, the IM group’s testosterone levels decreased significantly compared to the IC group (IM: mean ± SD = 3.23 ± 0.49 vs. IC: 5.20 ± 0.22 ng/mL, *p =* 0.001), but remained significantly higher than the SC group (*p* < 0.0001). Following a booster immunization at week 4, the IM group’s testosterone levels further decreased, reaching levels similar to the SC group by week 6 (IM: mean ± SD = 0.80 ± 0.18 vs. SC: 0.39 ± 0.05 ng/mL, *p =* 0.543). Throughout the remainder of the experiment until its conclusion, both IM and SC groups maintained consistently low testosterone levels, significantly lower than the IC group (*p* < 0.001). These results indicate that GnRH immunocastration can achieve testosterone reduction effects similar to surgical castration (Figure 2B).

After second immunization, the testes of the GnRH immunized group (IM) showed significant atrophy (Figure 3A). The testicular weight of the IM group was significantly lower than that of the intact control (IC) group (t (10) = 37.381, *p* < 0.0001), accounting for only 23.23% of the IC group (IM: mean ± SD = 0.39 ± 0.03 g; IC: mean ± SD = 1.67 ± 0.08 g). Similarly, the testicular volume of the IM group was significantly reduced (t (10) = 21.188, *p* < 0.0001), reaching approximately 25.38% of the IC group (IM: mean ± SD = 330.16 ± 32.01 mm^3^; IC: mean ± SD = 1301.08 ± 107.58 mm^3^) (Figure 3A). Furthermore, the testicular coefficient of the IM group was significantly lower than that of the IC group (t (10) = 33.564, *p* < 0.0001), representing only 23.17% of the IC group value (IM: mean ± SD = 0.08 ± 0.48%; IC: mean ± SD = 0.35 ± 1.91%) (Figure 3B). These results indicate that GnRH immunization significantly inhibited testicular development, leading to a marked decrease in testicular weight, volume, and coefficient.

Initially, there were no significant differences in the body weights of the three groups of rats (IC: mean ± SD = 200.42 ± 4.68 g vs. IM: mean ± SD = 197.13 ± 8.52 g vs. SC: mean ± SD = 198.82 ± 5.77 g; *p =* 1.000). After 2 weeks of treatment, the body weight of the surgical castration group (SC) was significantly lower than that of the intact control group (IC: mean ± SD = 225.33 ± 18.60 g vs. 268.78 ± 20.94 g; *p =* 0.008) and the immunocastration group (IM: mean ± SD = 225.33 ± 18.60 g vs. 270.02 ± 23.13 g; *p =* 0.007). At 4 weeks, the difference in body weight between the SC and IC groups was not significant (4 weeks: SC: mean ± SD = 300.50 ± 24.05 g vs. IC: mean ± SD = 324.28 ± 21.39 g; *p* = 0.216). This lack of significance may indicate a stabilization in the SC group’s weight as they adapted to the post-surgery environment. However, at 6 weeks, the SC group again showed a significant difference compared to the IC group (6 weeks: SC: mean ± SD = 343.20 ± 25.37 g vs. IC: mean ± SD = 382.57 ± 21.93 g; *p =* 0.031). However, over time, the weight difference between the SC and IC groups diminished, and by the 8th week, the difference was no longer significant (SC: mean ± SD = 398.38 ± 26.93 g vs. IC: mean ± SD = 422.67 ± 13.68 g; *p =* 0.159). In contrast, the immunocastration group did not show significant weight changes compared to the control group throughout the experimental period (*p =* 1.000), indicating no obvious short-term growth inhibition. This suggests that, unlike surgical castration, immunocastration does not induce a significant stress response (Figure 3C).

HE staining of the testicular tissue in the GnRH immunized group showed that the area of the seminiferous tubules was significantly reduced compared to the intact control group (Figure 3D,F), and the structure and morphology of the seminiferous tubules underwent substantial changes (Figure 3E,G). In the immunized group, the germ cells in the seminiferous tubules were primarily at the primary spermatocyte stage, with no round-headed or tadpole-shaped sperm observed, and the central vacuolization of the seminiferous tubules indicated a complete suppression of spermatogenesis (Figure 3G).

In the 12th week, we evaluated the sexual behavior of both immunized and control group rats, focusing on seven critical parameters. The results demonstrated that immunocastrated rats still displayed mounting behavior, with no statistically significant differences in mount latency and frequency compared to the control group (*p =* 0.631 and *p =* 0.16, respectively). However, for indicators more directly related to sexual behavior, such as intromission and ejaculation, the immunized rats did not exhibit these behaviors, showing a significant difference from the control group (*p =* 0.002). This suggests that while immunocastrated rats maintain a certain level of sexual motivation in the initial phase of sexual behavior (mounting), their ability to engage in intromission and ejaculation is significantly suppressed (Table 2).

These findings underscore the high efficacy of vaccine-induced immunocastration. These results indicated that this study successfully established an immunocastration animal model with castration effects similar to surgical castration, providing a foundation for evaluating the behavioral impacts of vaccine immunocastration in animals.

### 3.2. Effects of Castration Methods on Spatial Learning and Memory

Based on the observed impact of different castration methods on animal sexual behavior, the potential effects on other behavioral aspects, such as learning and cognition, remain unknown and were the focus of our study. To investigate this, we conducted a Morris Water Maze (MWM) test to assess the learning and cognitive abilities of rats subjected to different castration methods compared to a control group.

The Morris Water Maze (MWM) test consisted of two phases: the Spatial Navigation Test and the Probe Trial. During the Spatial Navigation Test, the progression of days significantly affected both escape latency (F(1.263, 18.950) = 169.397, *p* < 0.001) and total distance traveled (F(1.241, 18.611) = 107.169, *p* < 0.001), indicating that participants demonstrated significant learning progress throughout the experiment, reducing the time required to locate the target and optimizing their path selection. However, interaction analysis between time and group did not reveal significant group differences in escape latency (F(2.527, 38.900) = 0.621, *p =* 0.584) or total distance traveled (F(2.481, 37.222) = 1.879, *p =* 0.175). Further analysis of group effects confirmed that the IC, IM, and SC groups showed no significant differences in average escape latency (F(2, 15) = 0.701, *p =* 0.512) or total distance traveled (F(2, 15) = 1.199, *p =* 0.329), suggesting similar learning efficiency and path planning abilities across the three groups (Figure 4A,B).

During the Probe Trial, no significant differences were observed between groups in platform crossings (IC: mean ± SD = 9.50 ± 2.88; IM: mean ± SD = 11.83 ± 4.40; SC: mean ± SD = 8.50 ± 4.37; F(2, 15) = 1.126, *p =* 0.35), swimming velocity (IC: mean ± SD = 26.46 ± 3.53; IM: mean ± SD = 27.72 ± 7.47; SC: mean ± SD = 22.32 ± 3.05; F(2, 15) = 1.850, *p =* 0.191), or time spent in the platform zone (IC: mean ± SD = 4.80 ± 2.91; IM: mean ± SD = 6.03 ± 2.14; SC: mean ± SD = 5.69 ± 2.65; F(2, 15) = 0.366, *p =* 0.700). (Figure 4C–E). These results indicate no significant differences in exploratory behavior and motor abilities between groups, further supporting the conclusion that both surgical and immunological castration do not negatively impact animal learning and cognitive abilities. Additionally, the swimming paths of rats in the water maze were analyzed, further demonstrating the similarity in learning paths and patterns among the groups (Figure 4F). These findings contribute to the growing body of evidence supporting the feasibility and safety of GnRH immunocastration as an animal-friendly castration method.

### 3.3. Effects of Castration Methods on Emotional Behavior

The future trend in animal production emphasizes a more humane approach that considers the ethical needs of animals. Both immunocastration and surgical castration alter the levels of sex hormones within animals. However, it remains unclear whether these hormonal changes could induce further endocrine alterations, leading to emotional changes. In this study, we used the Sucrose Preference Test (SPT) and the Open Field Test (OFT) to assess the emotional behavior of three groups of rats.

The sucrose preference test revealed significant differences among treatment groups (Welch’s F(2, 9.166) = 22.382, *p* < 0.001). Multiple comparisons analysis demonstrated that the surgical castration SC group exhibited significantly lower sucrose preference (mean ± SD = 80.42% ± 5.21%) compared to both the intact control IC group (mean ± SD = 95.97% ± 1.78%, *p =* 0.001) and the immunological castration IM group (mean ± SD = 93.84% ± 2.10%, *p =* 0.002). Notably, no statistically significant difference was observed between the IC and IM groups (*p =* 0.227). These findings suggest that surgical castration may induce anhedonia-like behavior in rats, whereas immunological castration appears to have no significant impact on sucrose preference(Figure 5A).

In the Open Field Test (OFT), we compared the behavioral patterns of rats in the surgical castration (SC) group, the immunocastration (IM) group, and the intact control (IC) group (Figure 5B–G). ANOVA results indicated that the total movement distance in the SC group (mean ± SD = 1312.51 ± 514.98 cm) was significantly lower than the IM group (mean ± SD = 2447.68 ± 781.53 cm, *p =* 0.025) and not significantly different from the IC group (mean ± SD = 2316.52 ± 618.35 cm, *p =* 0.051). There was no significant difference between the IC and IM groups (*p =* 1.000) (Figure 5B). Furthermore, Kruskal–Wallis test results for central zone-related indicators showed significant differences among the groups in terms of central zone movement distance (H (2) = 11.843, *p =* 0.003), central zone duration (H (2) = 12.389, *p =* 0.002), central zone movement distance percentage (H (2) = 11.904, *p =* 0.003), and central zone duration percentage (H (2) = 12.389, *p =* 0.002) (Figure 5C–F). Specifically, SC rats tended to linger at the edges or remain stationary in the corners of the open field, rarely exploring the center, while IM and IC rats moved freely and explored the center (Figure 5G). These results suggest that SC rats exhibit significant anxiety and depression-like behaviors, whereas IM rats display behavioral patterns similar to those of IC rats. This indicates that surgical castration induces anxiety and depression-like behaviors, whereas immunocastration does not.

### 3.4. Effects of Castration Methods on Sociability and Social Novelty Preference

In the sociability test, the three groups of rats exhibited distinct spatial preferences and social behaviors. Surgically castrated (SC) rats displayed a strong preference for the familiar center zone (H (2) = 11.368, *p =* 0.003), spending significantly more time there (mean ± SD = 225.14 ± 59.54) compared to the IC (mean ± SD = 56.13 ± 43.11, *p =* 0.011) and IM (mean ± SD = 56.48 ± 30.86, *p =* 0.011) groups. Conversely, SC rats spent considerably less time in both the stranger chamber and the empty chamber than the IC groups (stranger chamber: SC mean ± SD vs. IC mean ± SD = 27.01 ± 40.99 vs. 177.32 ± 44.60, *p =* 0.024; empty chamber: SC mean ± SD vs. IC mean ± SD = 13.51 ± 16.64 vs. 66.39 ± 14.38, *p =* 0.008). No significant differences were found between the IC and IM groups in terms of time spent in any of the chambers(Figure 6A). These findings suggest that, compared to the control groups, surgical castration reduces social motivation and exploration, leading to an increased preference for familiar surroundings. Immunocastration, on the other hand, did not appear to have such an effect.

In the second phase of the social preference test, distinct behavioral patterns across groups were revealed. In the center area, rats from the surgical castration (SC) group exhibited significantly longer activity time (mean ± SD = 255.32 ± 51.38s) compared to the IM group (mean ± SD = 61.19 ± 25.44s, *p* < 0.0001) and the IC group (mean ± SD = 70.36 ± 33.15 s, *p* < 0.0001) (F(2,15) = 49.231, *p* < 0.0001).

Meanwhile, in areas related to social interaction, the SC group exhibited markedly different behavioral patterns from the other two groups. In the area containing a familiar rat (the familiar area), the activity time of the SC group (mean ± SD = 32.43 ± 36.94 s) was significantly lower than that of the IC group (mean ± SD = 141.84 ± 33.80 s, *p* < 0.0001) and the IM group (mean ± SD = 160.77 ± 28.59 s, *p* < 0.0001). Similarly, in the familiar area, the activity time of the SC group (mean ± SD = 12.24 ± 29.97 s) was also significantly lower than that of the IC group (mean ± SD = 87.77 ± 21.06 s, *p =* 0.014), but the difference between the SC group and the IM group did not reach statistical significance (IM: mean ± SD = 78.03 ± 19.20 s, *p =* 0.117). These results suggest that the interest of the SC group rats in social interaction may have been generally reduced, regardless of whether it was with unfamiliar or familiar conspecifics. It is noteworthy that there was no significant difference in activity time between the IC group and the IM group in all areas (*p >* 0.05), indicating that immunocastration had a relatively small impact on the social behavior and activity patterns of rats (Figure 6B).Through observation of rat trajectory paths, we found that the total movement distance during the acclimation period, the social test, and the social preference test was significantly lower in the SC group than in the IC and IM groups (Figure 6C).

## 4. Discussion

Animal welfare is a globally important issue, and all animals, including domestic and wild species, should receive appropriate protection and care. Many regions have implemented stringent regulations for surgical neutering, including mandatory anesthesia, veterinary supervision, and other safety protocols [27,28]. However, economically disadvantaged areas often lack such regulations due to financial limitations. In these regions, rudimentary neutering methods are commonly employed. Even in places where pain management and veterinary care are available, surgical neutering still carries inherent risks, such as postoperative pain and infection. Additionally, these procedures are complex, leading to increased labor costs, and postoperative care, particularly for wildlife, can be challenging to manage. In contrast, immunocastration presents a more humane and cost-effective alternative. It mitigates the risks and complications associated with invasive surgery and is more accessible and practical, especially in resource-constrained areas. As such, immunocastration holds great promise as a solution for improving animal welfare standards, offering a safer and more sustainable approach to population control.

In this study, we evaluated the feasibility and safety of a novel GnRH gene-engineered vaccine as a non-surgical castration strategy in male SD rats, focusing on its effects on reproductive physiology and various behavioral parameters. The vaccine successfully suppressed male reproductive functions, significantly increasing GnRH antibody levels, and reducing both testosterone levels and testis volume. It also effectively controlled sexual behavior, aligning with observations in different species [29], further affirming the efficacy of GnRH vaccines in reproductive regulation. Importantly, unlike the surgical castration group, which exhibited anxiety, depressive behaviors, and social interaction deficits, rats undergoing immunocastration showed stable performance in learning, emotional states, and social interactions, with no significant differences from the untreated control group. This suggests that the reproductive suppression induced by the GnRH vaccine may have a unique advantage in maintaining overall behavioral health.

Testosterone acts directly on key brain areas like the hippocampus and prefrontal cortex, influencing cognitive functions and emotional states [14,30]. Its reduction can lead to neurotransmitter imbalances, impaired neuroplasticity, and dysregulated stress responses, potentially causing anxiety, spatial memory impairments, and depressive behaviors [31,32]. Despite both immunocastration and surgical castration significantly reducing testosterone levels, only the immunocastrated rats did not exhibit depressive-like behaviors or social deficits. Several hypotheses might explain these differences:

Stress response variations: Surgical castration involves pain, trauma, and loss of bodily integrity, possibly triggering a significant psychological stress response, activating the hypothalamic-pituitary-adrenal (HPA) axis, and promoting the release of stress hormones like cortisol, which can negatively impact the brain’s reward system and emotional regulation [33,34,35]. In contrast, GnRH immunocastration is achieved by injecting a GnRH vaccine, inducing an autoantibody response against GnRH, blocking the interaction of endogenous GnRH with its receptor, thus suppressing the pituitary-gonadal axis function in a more gradual and non-invasive manner, reducing the acute stress burden, and favoring the maintenance of normal emotional and social behaviors.

Neuroplasticity and hormonal compensation: Surgical castration leads to a sudden drop in testosterone levels, which could dramatically affect brain structures sensitive to androgens, such as the hippocampus and prefrontal cortex. This may cause neural circuit remodeling and neurotransmitter imbalances, triggering depression, anxiety, and social deficits [36,37,38]. In contrast, GnRH immunocastration gradually lowers testosterone levels, giving the brain more time to adapt to hormonal changes, potentially alleviating or offsetting adverse effects through mechanisms of neuroplasticity, such as neurogenesis and synaptic remodeling [39]. Additionally, GnRH immunocastration might trigger compensatory hormonal responses, possibly increasing aromatase activity in brain areas like the hypothalamus, hippocampus, and amygdala, leading to elevated local estrogen levels that help mitigate anxiety and depression [40].

Moreover, while the surgically castrated rats displayed significant emotional and social behavior impairments, they showed no impact on reference memory in the Morris water maze test. This could be because the strong motivation to find the platform to avoid discomfort in water masked the cognitive effects of decreased testosterone levels. Although stress might reduce general behavioral motivation, this basic drive remains intact, pushing rats to complete spatial learning tasks [41,42]. Additionally, emotional, anxiety-related, and social behaviors depend primarily on brain regions like the amygdala and prefrontal cortex, which are particularly sensitive to stress and emotional changes [43]. In contrast, spatial memory relies mainly on the hippocampus, which, although affected by stress hormones like cortisol, can partially recover once stress is relieved, aiding the formation and consolidation of spatial memory, thereby offsetting the acute effects of surgical castration [33,44].

Our findings support the use of the GnRH vaccine as a non-invasive, progressive method of reproductive suppression, with potential advantages in avoiding behavioral and emotional disorders. This supports the use of the GnRH vaccine as a more humane and potentially less side-effect-prone strategy for fertility control, especially in situations where long-term suppression of reproductive functions is desired with minimal behavioral and psychological disturbances. However, this study has some limitations. First, the observation period was relatively short, and the long-term behavioral impacts of the GnRH vaccine require further research. Second, the small sample size means the reliability of the results needs to be verified in a larger sample. Third, our study did not involve other types of behavioral tests, such as fear memory and working memory, so a comprehensive assessment of the effects of the GnRH vaccine on cognition and emotion requires more behavioral experiments. Last, this study did not explore in depth the specific mechanisms by which the GnRH vaccine affects behavior, which should be investigated further using brain science techniques.

Future research should further explore the long-term effects of the GnRH vaccine and its impact on different types and genders of animals. Additionally, studying the relationship between hormone downregulation and behavioral and emotional changes, as well as the roles of hormonal compensation mechanisms and neuroplasticity in this process, will help to fully understand the biological and behavioral impacts of the GnRH vaccine in animal models, providing a scientific basis for future clinical applications.

## 5. Conclusions

Our study demonstrates that GnRH immunocastration achieves reproductive suppression comparable to surgical castration without adversely affecting the mood and behavior of SD rats. These findings provide preliminary but significant insights into the effects of the GnRH vaccine on the behavior of male SD rats, offering valuable implications not only for the welfare of experimental animals but also as critical reference information for fields such as pet management, livestock reproduction control, and wildlife conservation.

## Figures and Tables

**Figure 1 animals-14-02796-f001:**
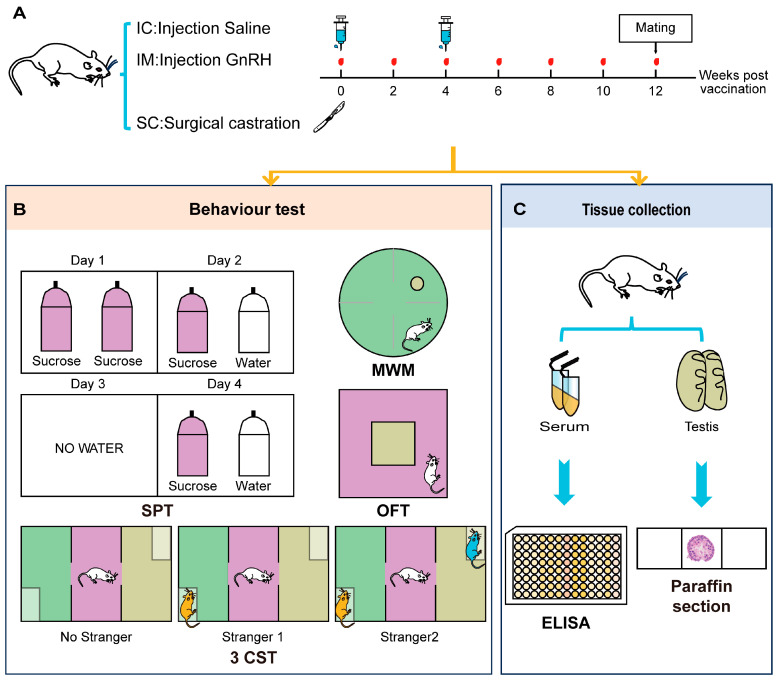
Experimental design and procedures for evaluating the effects of different treatments on rat behaviors. (**A**) Treatment Schedule: Rats were divided into three groups: IC (Injection Saline), IM (Injection GnRH), and SC (Surgical castration). Blood samples were collected at weeks 0, 2, 4, 6, 8, 10, and 12. Mating occurred at week 12. (**B**) Behavior Tests: SPT (Sucrose Preference Test): Conducted over four days. On Day 1, rats had access to two bottles of sucrose. On Day 2, rats had access to one bottles of sucrose and one bottles of water. On Day 3, no water was provided. On Day 4, rats had access to one bottle of sucrose and one bottle of water. MWM (Morris Water Maze): Rats were tested in a water maze to assess spatial learning and memory. OFT (Open Field Test): Rats were placed in an open field arena to assess general locomotor activity and anxiety-related behavior. 3CST (Three-Chamber Social Test): Conducted in three phases to assess social interaction. Phase 1: No stranger present. Phase 2: Stranger 1 introduced. Phase 3: Stranger 2 introduced. (**C**) Tissue Collection: At the end of the behavioral tests, rats were euthanized for tissue collection. Serum and testis samples were collected for further analysis. Serum samples were analyzed using ELISA, and testis samples were processed for paraffin sectioning and histological examination.

**Figure 2 animals-14-02796-f002:**
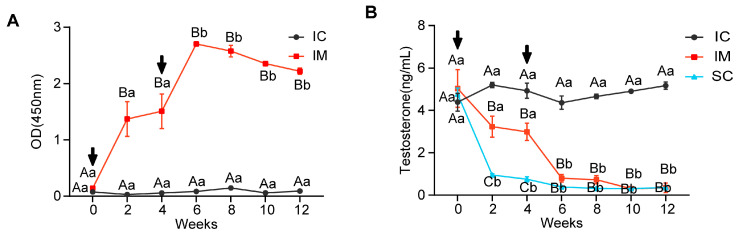
Changes in GnRH Antibody and Testosterone Levels in Immunized and Intact Control Groups. (**A**) Anti-GnRH antibody titers in serum samples from the IC and IM groups. (**B**) Testosterone levels in serum samples from the IC, IM, and SC groups. Statistical analysis was performed using two-way ANOVA. Data are presented as means ± SD. (**A**–**C**) Within a day, means without a common letter differed (*p* < 0.05). a,b Within a treatment group, means without a common letter differed (*p* < 0.05); Arrows indicate the timing of the primary vaccination and subsequent booster doses; OD = optical density.

**Figure 3 animals-14-02796-f003:**
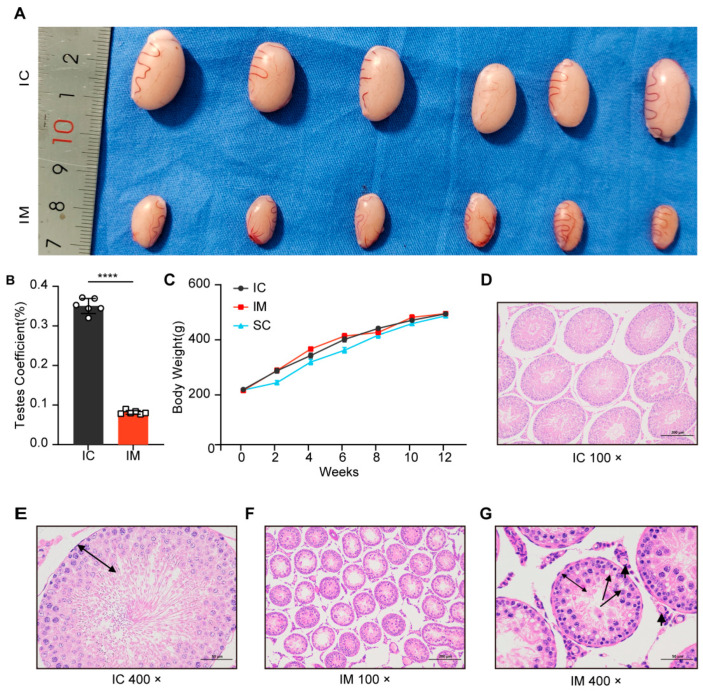
Effects of different treatments on testicular development and histology in rats. (**A**) Representative images of testes from rats treated with intact physiological saline control group (IC) and GnRH vaccine immunocastration group (IM) injections. The testes from the IC group are visibly larger compared to those from the IM group. (**B**) Testes coefficient (testes weight/body weight ratio) expressed as a percentage for IC and IM groups. The IM group shows a significantly higher testes coefficient compared to the IC group (**** *p* < 0.0001). (**C**) Growth curve showing body weight (g) of rats over a 12-week period for IC, IM, and surgical castration group (SC) groups. Data are presented as mean ± SD. (**D**) Histological section of testes from the IC group at 100× magnification, showing normal seminiferous tubule structure. (**E**) Histological section of testes from the IC group at 400× magnification. (**F**) Histological section of testes from the IM group at 100× magnification, showing atrophic seminiferous tubule structure. (**G**) Histological section of testes from the IM group at 400× magnification, cavitation in spermatocyte (long arrows) and Leydig cells (short arrows) were observed, germinal epithelium height decreased (double side black arrow).

**Figure 4 animals-14-02796-f004:**
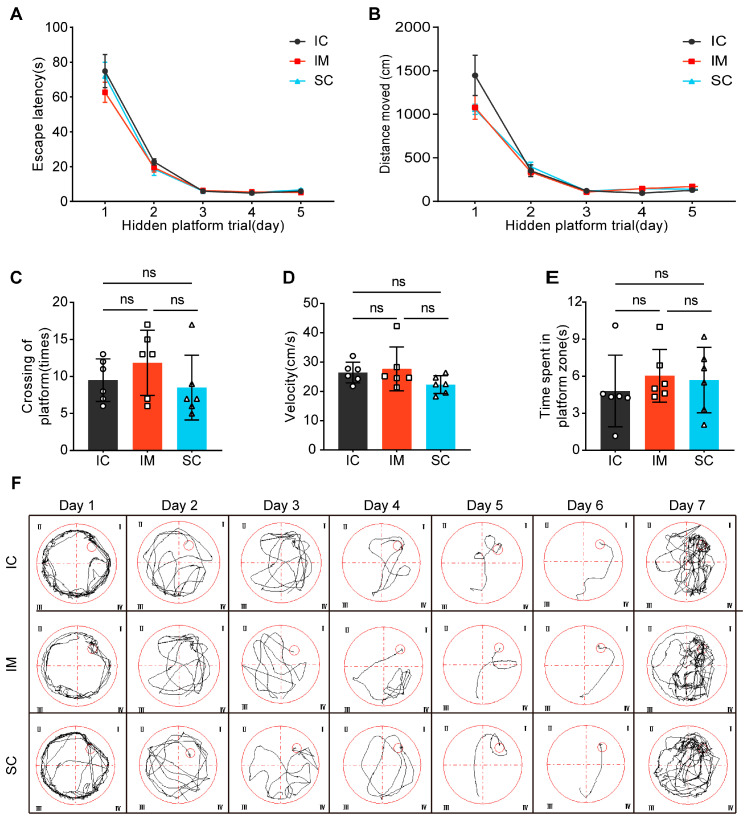
The effects of GnRH immunocastration and surgical castration on spatial learning in male Sprague-Dawley rats. (**A**) Escape latency: The time it took for the rats to find the hidden platform. (**B**) Total movement distance: The total distance traveled by the rats during the trial. (**C**) The number of platform crossings: The number of times the rats crossed the location of the hidden platform. (**D**) Velocity: The speed of the rats during the trial. (**E**) Time spent in the platform zone: The amount of time the rats spent in the zone where the hidden platform was previously located. (**F**) Swimming tracks: Representative swimming paths of the rats. Statistical analysis was performed using One-way ANOVA and Two-way ANOVA. Data are presented as means ± SD. Significance is indicated as follows: ns = not statistically significant.

**Figure 5 animals-14-02796-f005:**
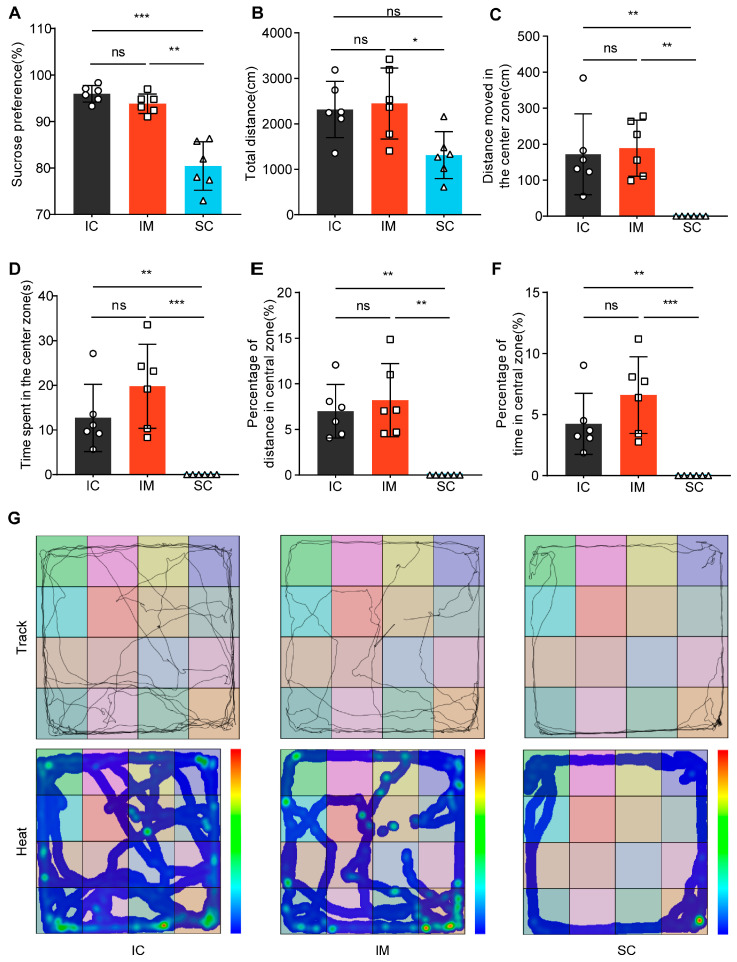
Differential effects of GnRH immunocastration and surgical castration on emotional behavior in male Sprague–Dawley rats. (**A**) In the Sucrose Preference Test (SPT), the SC group exhibited a decreased preference for sucrose. (**B**) In the Open Field Test (OFT), the SC group exhibited a shorter total distance moved than the IC group. (**C**) In the OFT, the SC group showed a shorter distance moved in the central zone compared to the IC group. (**D**) In the OFT, the time spent in the central area by the SC group was significantly lower than the control group. (**E**) In the OFT, the proportion of distance traveled in the central area was significantly lower in the surgically castrated group compared to the IC group. (**F**) In the OFT, the proportion of time spent in the central area was significantly lower in the SC group than in the IC group. (**G**) Trajectory path and heatmaps across the three stages of the OFT. Statistical analysis was performed using One-way ANOVA. Data are presented as means ± SD. Significance is indicated as follows: * *p* < 0.05; ** *p* < 0.01; *** *p* < 0.001; ns = not statistically significant; SPT = Sucrose Preference Test; OFT = Open Field Test.

**Figure 6 animals-14-02796-f006:**
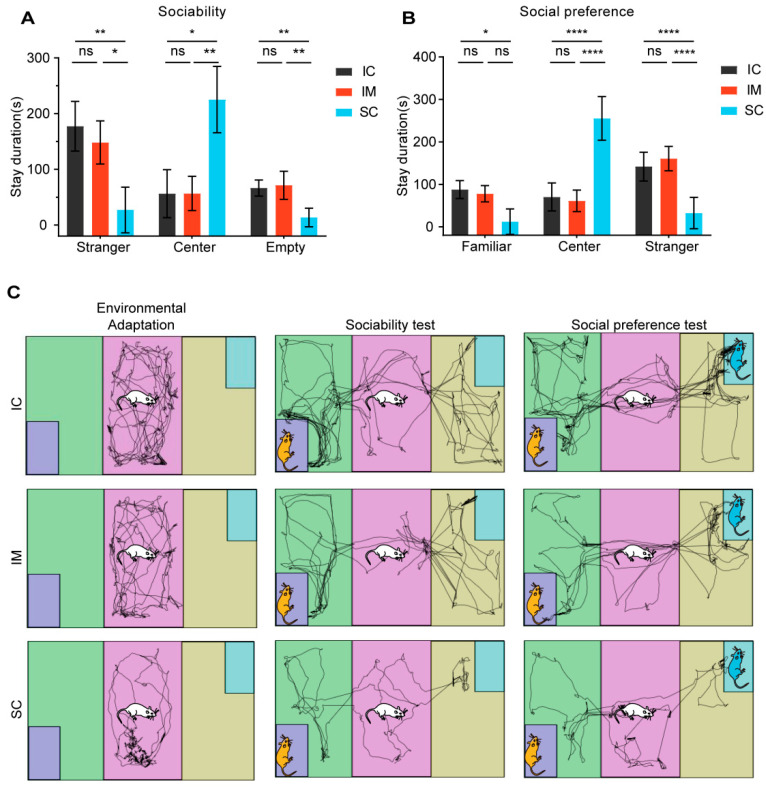
The effects of GnRH immunocastration and surgical castration on sociability and social novelty preference in male Sprague–Dawley rats. (**A**) In the sociability phase, the SC group overall spent less time interacting with strangers compared to the IC group. (**B**) In the social novelty preference phase, the SC group overall spent less time interacting with novel conspecifics compared to the IC group. GnRH immunocastration did not affect sociability or social novelty preference in rats. (**C**) Trajectory path of the three stages of the 3-CST. Statistical analysis was performed using Two-way ANOVA. Data are presented as means ± SD. Significance is indicated as follows: * *p* < 0.05; ** *p* < 0.01; **** *p* < 0.0001; ns = not statistically significant; 3-CST = Three-chamber sociability test.

**Table 1 animals-14-02796-t001:** Copulation test outcome measure definitions.

Outcome Measure	Definition
Mount latency, ML	Time required for a male rat to mount after being placed in the same cage with a female rat for the first time
Mount frequency, MF	The total number of mounts by a male rat on a female rat within 30 min, regardless of intromission
Intromission latency, IL	Time required for a male rat to achieve the first intromission after being placed in the same cage with a female rat
Intromission frequency, IF	The total number of intromissions by a male rat on a female rat within 30 min
Ejaculation frequency, EF	The number of ejaculations by a male rat within 30 min
Postejaculatory interval, PEI	The time interval between each ejaculation and the next intromission behavior in rats
Intromission ratio, IR	Intromission ratio: Number of intromissions/(number of intromissions + number of mounts)

**Table 2 animals-14-02796-t002:** Statistical analysis of male rat sexual behavior of IM and IC groups.

Variables	Groups		t(df)/Z-Value	*p*-Value and Significance
	IC(*n* = 6)Mean ± SD/Median [IQR]	IM(*n* = 6)Mean ± SD/Median [IQR]		
Mount latency (s)	424.00 [155.25, 742.75]	367.50 [165.00, 1022.00]	−0.480	0.631 ns
Intromission Latency (s)	448.50 [223.50, 778.25]	1800.00 [1800.00, 1800.00]	−3.077	0.002 **
Ejaculation Latency (s)	4.00 [2.75, 5.25]	1800.00 [1800.00, 1800.00]	−3.083	0.002 **
Intromission Frequency (£)	10.00 [5.75, 12.25]	0.00 [0.00, 0.00]	−3.077	0.002 **
Mount Frequency (£)	7.67 ± 5.61	3.67 ± 2.8	t (7.353) = 1.562	0.16 ns
Ejaculation Frequency (£)	4.50 [2.75, 7.25]	0.00 [0.00, 0.00]	−3.083	0.002 **
Postejaculatory Interval (s)	104.50 [70.25, 151.50]	0.00 [0.00, 0.00]	−3.077	0.002 **
Intromission Ratio (%)	58.69 [44.82, 74.73]	0.00 [0.00, 0.00]	−3.077	0.002 **

The behavioral data of male rats in the IM and IC groups are presented as mean ± standard deviation (SD) for normally distributed data and median (interquartile range (IQR)) for non-normally distributed data. An unpaired Student’s *t*-test was used for comparisons of normally distributed data, and the Mann–Whitney U test was used for non-normally distributed data. Significance is indicated as follows: ** *p* < 0.01; s: Second, £: frequency.

## Data Availability

The data for this study are available upon request from the corresponding author (shangquangan@gdou.edu.cn). Data will be provided upon acceptance of the Terms of Use.

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
