# Peer review of "Behavioral Assessment Reveals GnRH Immunocastration as a Better Alternative to Surgical Castration"

_animals, 2024, doi:10.3390/ani14192796_

Round 1

Reviewer 1 Report

Comments and Suggestions for Authors

The article is scientifically of high quality and it is an important contribution to the public discussion about whether immunocastration is in the light of animal welfare superior to sugical castration. This question has not been sufficiently answered especially in pig production. Animal welfare groups around the world argue that immunocastartion of piglets is the most animal friendly method to suppress boar taint in pork. But still there is a hesitation in the porok industry to apply the immunocastraion as the golden standard in terms of animal wellbeing. Insofar, the results and the conclusions of the article is very important to scientifically underline the demands of the animal welfare activists to implement the immunocastration in the field.

The reviewer has one question: why did the authors the GnRH vaccine genetically optimize? The explanation "to improve the immunogenicity of the vaccine" is OK, but in the light of the question "is the GnRH vaccine influence behavioral attitudes of the rats" wouldn't it be more logic to use the vaccine as it is commercially available. The animal welfare question should rather be answered for the commercial vaccine.

It is no demand, but the reviewer thinks that two papers would be better: 1. one paper on the impact of the immunocatstration with a commercial vaccine on behavioral chracteristics, and 2. one paper on the sophisticated geneticingeneering of the commercial vaccine to improve its immunogenicity.

At least the authors should explain in the introduction, why they genetically engeneered the commercial vaccine, and that they are convinced that there is no difference in the characteristics of the genetically engeneered vaccine compared to the commercial vaccine except of the immunogenicity. This would be imporrtant since many a reader will ask himself or herself, whether the results do really reflect the impact of the commercial vaccine on the behavioral chracteristics of immunosized rats.  

Comments on the Quality of English Language

Minor amandments seem to be necessary.

E.g.:  1st sentence of introduction: "...control is crucial manipulate in fields..." is linguistically inadaquate

Reviewer 2 Report

Comments and Suggestions for Authors

Revision

In light of the need to subject animals to castration within the context of livestock reproduction control and wildlife conservation, this manuscript aims to evaluate the different behaviors of animals (rats) subjected to surgical castration and GnRH immunocastration.

It is generally concluded that GnRH immunocastration is a superior alternative to surgical castration, as it promotes greater animal welfare, providing a more humane solution.

The reading of the manuscript raised several questions that we consider relevant, which I will now outline:

1.     1. This manuscript titled "Behavioral Assessment Reveals GnRH Immunocastration as a Better Alternative to Surgical Castration" aims to contribute to a better understanding of the advantages of using immunocastration methodologies in livestock reproduction control and wildlife conservation.

- We are of the opinion that this issue should be more thoroughly detailed, and it should be mentioned to what extent male castration is an important management tool in livestock management..

- It should be mentioned to what extent castration can be an important measure in terms of wildlife conservation, as this aspect is not clearly addressed in the manuscript.

2.     2. The issue of boar taint is a problem that affects the production of both domestic and wild pigs, but it does not impact the production of all meat-producing species, whether domestic or wild.

- We believe that a clear reference to the specific species that are castrated during their production period should be included, as this is not clarified in the text.

- The description of the cause of "boar taint" in non-castrated animals seems incomplete, as the entire study aims to draw an analogy between what occurs in rats and in pigs (where castration is justified for production purposes). It should be noted that in these animals, boar taint is due to the accumulation of both androstenone and skatole in edible tissues. We believe this explanation should be presented with greater precision.

3.     In our view, the term "meat quality" is sometimes used incorrectly.

- "Meat quality" is a broad term that encompasses a wide range of characteristics of meat, including physical-chemical and organoleptic attributes, among others. Therefore, we believe this term should be used more precisely.

- It is important for the authors to recognize that while castration can improve certain meat quality characteristics (such as odor), others may remain unchanged or be adversely affected. This specifically pertains to pork production.

4.      Materials and Methods (Surgical Procedures)

- In production animals, the methodology for surgical castration, particularly in pigs older than 7 days, is regulated by European legislation and includes analgesia and veterinary care to prevent inflammatory processes. When these regulations are adhered to, the animal's suffering associated with the surgical procedure is minimized due to the implementation of these measures.

In the manuscript, these measures are not mentioned, which undermines the comparison with the castration of meat production animals.

Line 125 - “Rats in this group underwent bilateral orchiectomy under anesthesia induced with 125 sodium pentobarbital (50 mg/kg, intraperitoneal injection) at the onset of the study (0 week).”- Nothing is mentioned regarding analgesia or other types of post- surgical care.

5.     Materials and Methods

The type of methodology used for Behavioral Assessment is very interesting:

Line 74 - A meticulously designed experimentsuite was employed, encompassing the Morris water maze to assess cognitive functions related to learning and memory, the sucrose preference test and the open field test for evaluating affective states, and the three-chamber social interaction test (3 CST) tomonitor social behaviors

- However, we believe that the manuscript does not adequately explain how the results of these tests are relevant to the realities of animal production. The lack of insight into these aspects throughout the results and discussion sections leads us to find some of these results lacking in significance (interest).

The data treatment is correct and appropriate for the results obtained.

We believe that a new approach to the results, taking into account the considerations made, would significantly improve the manuscript.

Comments on the Quality of English Language

Moderate editing English language required.

Reviewer 3 Report

Comments and Suggestions for Authors

By and large, this is a very well-designed and well-executed study comparing some physiological and behavioral effects of a GnRH-based immunocontraceptive vaccine on rats.  With some glitches (below) the results are very clearly presented. As the authors note, the sample size is a bit small, but the effects observed are dramatic enough that it doesn't much matter.

My substantive comments are mostly minor.

First, the simple summary and the abstract both start off framing the study in the context of use of castration to improve the taste of meat in livestock production.  However, the text of the article makes almost no mention of this application in either the introduction or the discussion.  Either this framing should be removed from the summary and abstract, or the topic needs to be considered more seriously in the text, and the authors should evaluate the relevance of this work to use in livestock.

This may seem petty, but the abbreviations used to designate the three treatment groups are very confusing and non-intuitive.  (Both the "I" and the "C" stand for different things in different abbreviations.  Should IC stand for "intact control" or Immunocastration?)  I had to keep looking back in the text to see what they stood for; and the authors restated the definition of the abbreviations at several places in the text.  Please pick different abbreviations for the treatment groups whose meaning is more obvious and memorable.

TC ("Testicular coefficient") is a very simple concept. But so far as I can tell, the formula (l. 180) is wrong (should be Body weight in grams, not kg, yes?), and the values reported seem to be different in Fig. 3B and l. 335.  Please harmonize; this is an easy fix.

Methodologically, it is not really clear why the authors chose not to test sexual behavior in surgically castrated rats.  It would have been helpful to have that baseline for comparison.  Maybe more explanation would clear up the decision.

Finally, the subtitles in the results section 3 are inconsistent and in some cases inappropriate.  Please just use the subtitles to briefly describe the topics covered in the section, rather than summarizing the results.

Very minor editorial comments and corrections:

l. 202 "waswas"

l. 241.  After "anxiety-related," something is missing.

l. 279-280.  First sentence of the paragraph is not needed.

l. 344.  The difference in body weights between SC and IC groups are not significant at 4 weeks; only at 2 and 6 weeks.

l. 390.  What does "time significantly affected" mean?  Does "time" refer to the day of the trial?

l. 406-408.  If the ANOVA's are n.s., omit the Bonferroni post hoc tests.

l. 411-412.  The sentence is missing something.

l. 441.  Rats, not mice.  (I accidentally switch my species out all the time.)

l. 475-477.  I don't think the first paragraph belongs in the Results section.

l. 501 "familiar..." rat area?  Something is left out.

l. 521-516.  Delete.  The first sentence belongs in the discussion, and the rest is the template instructions.

Fig. 6C.  What exactly do the lines represent?  Does each line represent the movements of a single rat, or is it some combination?

Round 2

Reviewer 2 Report

Comments and Suggestions for Authors

The manuscript presents in this second version sufficient quality to be published. The authors complied with the requested recommendations